

# Association between aberrant APC promoter methylation and breast cancer pathogenesis: a meta-analysis of 35 observational studies

Dan Zhou[1,2], Weiwei Tang[3], Wenyi Wang[3], Xiaoyan Pan[3], Han-Xiang An[3] and Yun Zhang[1,2]

[1] Department of Translational Medicine, Xiamen Institute of Rare Earth Materials, Xiamen, China
[2] Department of Translational Medicine, Key Laboratory of Design and Assembly of Functional Nanostructures, Fujian Provincial Key Laboratory of Nanomaterials, Fujian Institute of Research on the Structure of Matter, Fuzhou, China
[3] Department of Medical Oncology, The First Affiliated Hospital of Xiamen University, Xiamen, China

## ABSTRACT

**Background.** Adenomatous polyposis coli (APC) is widely known as an antagonist of the Wnt signaling pathway via the inactivation of $\beta$-catenin. An increasing number of studies have reported that APC methylation contributes to the predisposition to breast cancer (BC). However, recent studies have yielded conflicting results.

**Methods.** Herein, we systematically carried out a meta-analysis to assess the correlation between APC methylation and BC risk. Based on searches of the Cochrane Library, PubMed, Web of Science and Embase databases, the odds ratio (OR) with 95% confidence interval (CI) values were pooled and summarized.

**Results.** A total of 31 articles involving 35 observational studies with 2,483 cases and 1,218 controls met the inclusion criteria. The results demonstrated that the frequency of APC methylation was significantly higher in BC cases than controls under a random effect model (OR = 8.92, 95% CI [5.12–15.52]). Subgroup analysis further confirmed the reliable results, regardless of the sample types detected, methylation detection methods applied and different regions included. Interestingly, our results also showed that the frequency of APC methylation was significantly lower in early-stage BC patients than late-stage ones (OR = 0.62, 95% CI [0.42–0.93]).

**Conclusion.** APC methylation might play an indispensable role in the pathogenesis of BC and could be regarded as a potential biomarker for the diagnosis of BC.

## INTRODUCTION

Breast cancer (BC) is the most common malignancy and the leading cause of cancer death among females in both well and poorly developed countries, accounting for approximately 15% of all cancer deaths in 2012 (*Torre et al., 2015*). It is well established that BC is a clinically and pathologically heterogeneous disease and has been categorized into five subtypes (i.e., luminal A and B, human epidermal growth receptor-2, triple negative and basal-like) based on various biological markers (*Inoue & Fry, 2015*). Risk factors including

Corresponding authors
Han-Xiang An,
anhanxiang@yahoo.com
Yun Zhang, zhangy@fjirsm.ac.cn

reproductive, hormonal and environmental factors, have been associated with an increased incidence of BC (*Harrison et al., 2015*). Previous studies have reported that early detection using mammography is effective and can improve the overall survival rate (*Brooks et al., 2010*). However, false positive mammograms always result in the over-diagnosis and over-treatment of developing BC. Therefore, no acknowledged biomarker has yet been proven to be sufficiently sensitive and specific for routine use in clinical diagnosis.

Epigenetic as well as genetic alterations are both stable and heritable and occur in tumor suppressor genes involved in tumourigenesis. The most common epigenetic alteration involving aberrant DNA methylation, a reliable and sensitive biomarker for nearly all types of cancer including breast cancer, often leads to the transcriptional silencing of tumor suppressor genes (*Zmetakova et al., 2013*). Several studies have demonstrated that tumor DNA derived from malignant cells can be detected in various bodily fluids and serum of BC patients and can potentially serve as a non-invasive diagnostic material (*Martínez-Galán et al., 2014*). A growing number of tumor suppressor genes has been shown to be directly involved in cell cycle regulation, DNA repair, cell signal transduction and angiogenesis (*Dumitrescu, 2012*). Notably, the promoter methylation of genes involved in the canonical Wnt signaling pathway, which regulates cell differentiation, proliferation and homeostasis, are observed more often in BC patients compared with cancer-free controls (*Klarmann, Decker & Farrar, 2014*).

The adenomatous polyposis coli (APC) gene is widely known as an antagonist of the Wnt signaling pathway via the inactivation of $\beta$-catenin, which is regarded as a transcriptional activator (*Virmani et al., 2001*). The APC gene, located at chromosome 5q21–5q22, was originally implicated in colorectal cancer (*Van der Auwera et al., 2008*). The inhibition or down-regulation of APC expression through APC promoter methylation contributes to the formation of colorectal cancer (*Ashktorab et al., 2013*). Similar to the findings in colorectal cancer, APC promoter methylation is associated with various early- or late-stage human malignancies, including BC (*Matsuda et al., 2009*). The promoter hypermethylation of APC is most often related to the nuclear accumulation of $\beta$-catenin, which may result in the loss of cell growth control (*Sparks et al., 1998*). Thus, APC promoter methylation, which acts as a non-invasive biomarker, can be used to distinguish BC patients from cancer-free controls. However, recent studies have yielded conflicting results with regard to the significant association between APC methylation and BC pathogenesis. *Wojdacz et al.* (*2011a*) reported that there was no significant difference in the frequency of APC methylation in peripheral blood leukocyte DNA between BC patients and cancer-free controls. *Cho et al.* (*2010*) also showed that the APC gene was rarely hypermethylated in blood DNA in BC patients.

Given these controversial results, we conducted this comprehensive meta-analysis of the current observational studies to evaluate the association between the aberrant methylation of the APC promoter and increased BC risk.

## MATERIALS & METHODS

### Search strategy

Eligible studies were identified by searching the following databases until February 2016: the Cochrane Library, PubMed, Web of Science and Embase. No language restrictions or

lower data limits were imposed; only abstracts, unpublished and incomplete studies were excluded. Titles, abstracts of potential references and reference lists from relevant studies were carefully checked. We performed the search strategy using the following search terms and their various combinations: "APC," "Adenomatous polyposis coli," "methylation," "breast cancer," "breast neoplasm" and "mammary carcinoma."

## Selection criteria

The studies included in the present meta-analysis addressed the association between APC methylation and increased BC risk. Our inclusion criteria were as follows: (1) provided sufficient data on the frequency of APC methylation in BC patients and controls; (2) original observational studies in full-text form; and (3) when several studies overlapped, the most recent or large-scale article was selected. The following were exclusion criteria: (1) data based on reviews, animal models, case reports or cell line studies; (2) studies lacking key information necessary for calculations; (3) duplicated studies; and (4) studies including BC patients or controls who underwent radiotherapy and chemotherapy which may influence APC promoter methylation levels.

## Data extraction

The relevant data were extracted from the eligible studies independently by two authors (D Zhou and WW Tang). Differing opinions, if any, were resolved by discussion in accordance with the original literature. The following information was extracted in a predefined table: the name of the first author, the year of publication, the country of origin, the sample type, the experimental methods used to detect APC methylation, sample size, tumor stage, tumor grade and APC methylation frequencies. Additionally, we classified stage 0, I and II as early-stage BC and stage III and IV as late-stage BC, as confirmed by the AJCC staging system. Furthermore, grades I and II were combined as low-grade BC; grade III was regarded as high-grade BC. This meta-analysis was performed following the statement of preferred reporting items set by the PRISMA Group (File S1) (*Moher et al.*, *2009*).

## Statistical analysis

All analyses were carried out using Review Manager 5.3 (Cochrane Collaboration) and Stata 12.0 (Stata Corporation) software. Forest plots were designed to estimate relative study-specific effects according to the 95% confidence interval (CI). The association between APC promoter methylation and BC risk was evaluated by calculating the odds ratio (OR) with corresponding 95% CI. For individual studies the OR was represented by a square and the 95% CI by a horizontal line in the centre of the forest plot. The OR and associated 95% CIs in the centre of the forest plot were plotted on a logarithmic scale. When a CI did not include 1.0, a correlation was deemed statistically significant. Heterogeneity between the included studies was quantified through $Q$-tests based on the chi-square test and $I^2$ value. An $I^2$ value $>50\%$ and a $p < 0.10$ denoted strong heterogeneity, an $I^2$ value $= 25$–$50\%$ denoted a moderate degree of heterogeneity and an $I^2$ value $<25\%$ or a $p > 0.10$ denoted mild heterogeneity (*Higgins et al.*, *2003*). A random effect model was used when statistical heterogeneity existed among studies ($p < 0.1$). Otherwise, the fixed effect model was employed (*Li et al.*, *2014*). Moreover, the subgroup meta-analyses were also performed

according to region, experimental methods for detecting APC methylation, and sample types in order to explore the potential origin of inter-study heterogeneity. In addition, we conducted a sensitivity analysis by removing a single study to examine the stability of the results. The funnel plot, Begg's test and Egger's test were investigated in order to determine the degree of publication bias. The treatment effect was plotted against a measure of study size in the funnel plot. When publication bias was present, the shape of the funnel plot was asymmetric. Trim and fill analysis was used to estimate the number of potential missing studies resulting from the asymmetry of the funnel plot.

## RESULTS

### Study selection and characteristics

The selection process is displayed as a flow chart in Fig. 1 based on the search strategies as previously described. After a careful initial search of the abstracts, 74 potentially relevant articles were identified excluding 1 duplicate and 93 irrelevant studies. Then, we reviewed the full text articles. Among these studies, 43 were excluded (21 articles did not design a control group; 9 articles focused on BC cell lines; 8 articles lacked available data; and 5 articles were reviews). Finally, 31 studies published from 2001 to 2016 involving 35 studies were included in this systematic meta-analysis (PubMed 19, Web of Science 10, Embase 2).

The general characteristics of eligible studies were summarized and displayed in Table 1. A total of 2,483 BC patients and 1,218 controls were employed in multiple countries or regions including Asia (n = 10) (*Jin et al., 2001*; *Jing et al., 2010*; *Jung et al., 2013*; *Lee et al., 2004*; *Liu et al., 2007*; *Park et al., 2011b*; *Prasad et al., 2008*; *Zhang et al., 2007*), Europe (n = 13) (*Fridrichova et al., 2015*; *Hoque et al., 2009*; *Jeronimo et al., 2008*; *Martins et al., 2011*; *Matuschek et al., 2010*; *Muller et al., 2003*; *Parrella et al., 2004*; *Rykova et al., 2004*; *Van der Auwera et al., 2009a*; *Van der Auwera et al., 2009b*; *Van der Auwera et al., 2008*; *Wojdacz et al., 2011b*), Africa (n = 2) (*Hoque et al., 2006*; *Swellam et al., 2015*), North America (n = 9) (*Brooks et al., 2010*; *Chen et al., 2011*; *Cho et al., 2010*; *Dulaimi et al., 2004*; *Lewis et al., 2005*; *Shinozaki et al., 2005*; *Taback et al., 2006*; *Virmani et al., 2001*) and Oceania (n = 1) (*Pang et al., 2014*). Furthermore, the methylated APC levels in BC patients and controls were examined with 6 methods. Of these methods, methylation specific PCR (MSP) was adopted in 17 studies, quantitative real-time MSP (QMSP) was used in 9 studies, methylation specific-multiplex ligation-dependent probe amplification (MethyLight) was used in 4 studies, methylation specific-multiplex ligation-dependent probe amplification (MS-MLPA) was employed in 2 studies, methylation-sensitive high-resolution melting analysis (MS-HRM) was used in 2 studies and pyrosequencing was used in only 1 study. Furthermore, BC tissues (i.e., fresh frozen tissues, formalin fixed paraffin-embedded tissues and tissues from surgery), samples derived from blood (i.e., blood cells and serum) and needle aspirated fluid (NAF) were enrolled to assess the methylation levels of the APC promoter.

### Meta-analysis

The pooled results of this meta-analysis reflected the association between APC promoter methylation and BC pathogenesis (Fig. 2). Due to the existence of significant heterogeneity

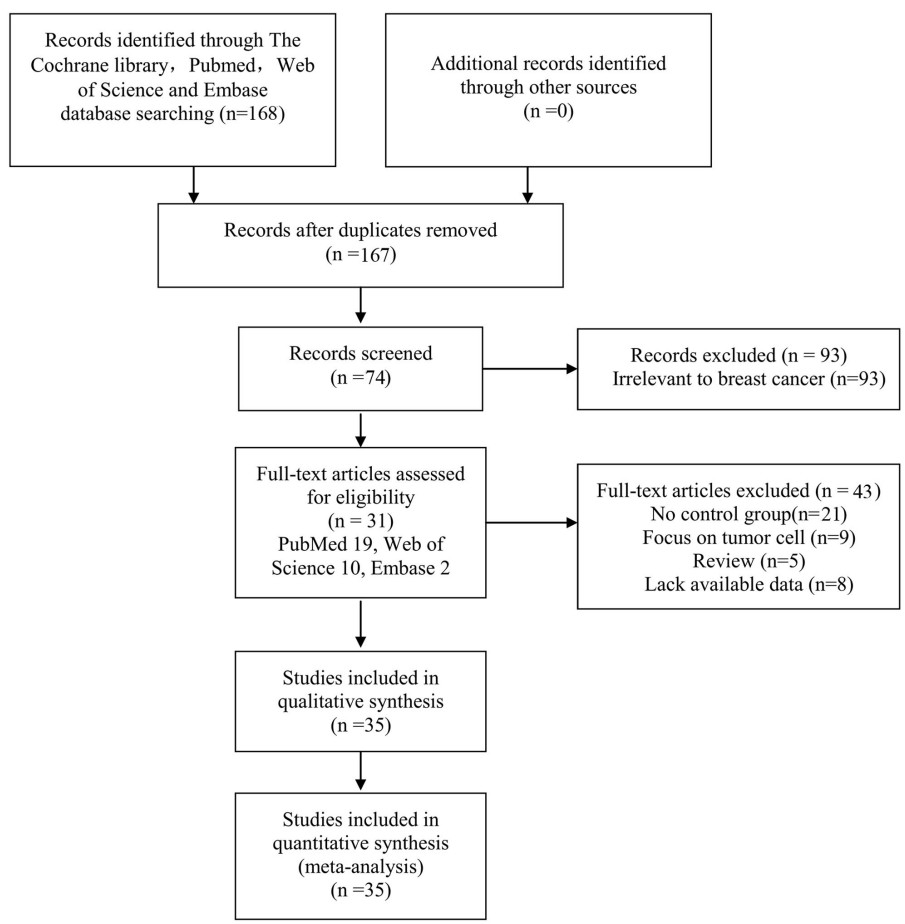

**Figure 1** **Flow chart of the collection of studies for this meta-analysis.**

among the included studies ($p < 0.00001$, $I^2 = 65\%$), the random effect model was adopted to evaluate the combined effects of APC promoter methylation. The overall analysis indicated that the frequency of APC promoter methylation was remarkably higher in BC patients than in cancer-free controls. The combined OR for 35 included relevant studies showed that APC methylation was significantly correlated with increased BC risk and the absence of APC expression played an important role in BC pathogenesis (OR = 8.92, 95% CI [5.12–15.52]).

## Sensitivity analysis

A sensitivity analysis was conducted by omitting one individual study every time to evaluate the stability of the pooled OR and to choose the heterogeneous study. As shown in Fig. 3, the combined OR between APC methylation and increased BC risk was indeed reliable without heterogeneous studies.

## Subgroup analysis

Due to the significant existence of inter-study heterogeneity ($p < 0.00001$, $I^2 = 65\%$), subgroup analysis based on region, experimental methods for the detection of APC methylation and sample types were carried out to appraise the sources of the heterogeneity

**Table 1  General characteristics of eligible studies.**

| Author | Year | County/region | Method | Sample type | M/N | | Stage (M/N) | | Grade (N/M) | |
|---|---|---|---|---|---|---|---|---|---|---|
| | | | | | BC | Control | Early | Late | Low | High |
| Brooks JD | 2010 | USA | QMSP | Serum | 1/50 | 6/148 | – | – | – | – |
| Chen KM | 2011 | USA | MS-MLPA | FFT | 12/17 | 1/10 | – | – | – | – |
| Cho YH | 2010 | USA | MethyLight | FFT | 21/40 | 12/27 | – | – | – | – |
| Dulaimi E 1 | 2004 | USA | MSP | Surgery | 15/34 | 0/6 | 14/29 | 1/5 | 6/13 | 9/18 |
| Dulaimi E 2 | | | | Serum | 10/34 | 0/20 | 9/29 | 1/5 | 4/13 | 6/18 |
| Fridrichova I | 2015 | Slovak Republic | Pyro | FFPET | 144/206 | 0/9 | – | – | – | – |
| Hoque MO | 2006 | West Africa | QMSP | Blood | 8/47 | 0/38 | – | – | – | – |
| Hoque MO | 2009 | Italy | QMSP | FFPET | 56/112 | 3/32 | – | – | – | – |
| Jeronimo C | 2008 | Portugal | QMSP | FFPET | 55/66 | 10/12 | – | – | – | – |
| Jin Z | 2001 | Japan | MSP | Surgery | 18/50 | 0/21 | 13/36 | 4/10 | – | – |
| Jing F | 2010 | China | MSP | Serum | 14/50 | 0/50 | – | – | 7/25 | 12/25 |
| Jung EJ | 2013 | Korea | MS-MLPA | Surgery | 19/60 | 0/60 | 17/53 | 2/7 | 13/40 | 6/20 |
| Lee A | 2004 | Korea | MSP | NAF | 14/33 | 0/19 | 13/31 | 1/2 | – | – |
| Lewis CM | 2005 | USA | MSP | NAF | 15/27 | 14/55 | – | – | – | – |
| Liu Z | 2007 | China | MSP | Surgery | 28/76 | 0/76 | 15/54 | 13/22 | 15/48 | 13/28 |
| Martins AT | 2011 | Portugal | QMSP | NAF | 144/178 | 18/33 | – | – | – | – |
| Matuschek C | 2010 | Germany | MethyLight | Blood | 25/85 | 2/22 | 5/42 | 16/35 | – | – |
| Müller HM | 2003 | Austria | MethyLight | Serum | 6/26 | 0/10 | – | – | – | – |
| Pang JM | 2014 | Australia | MS-HRM | FFPET | 39/80 | 0/15 | – | – | – | – |
| Park SY | 2011 | South Korea | MethyLight | FFPET | 31/85 | 2/30 | – | – | 13/30 | 6/20 |
| Parrella P | 2004 | Italy | MSP | Tissue | 15/54 | 1/10 | – | – | – | – |
| Prasad CP 1 | 2008 | India | MSP | Surgery | 6/32 | 0/5 | 2/19 | 4/13 | – | – |
| Prasad CP 2 | | | | Serum | 11/50 | 0/50 | – | – | 4/28 | 7/22 |
| Rykova EY | 2004 | Russia | MSP | Blood | 4/10 | 0/6 | – | – | – | – |
| Shinozaki M | 2005 | USA | MSP | FFPET | 74/151 | 0/10 | – | – | – | – |
| Swellam M | 2015 | Egypt | MSP | Serum | 113/121 | 0/66 | 81/86 | 32/35 | 84/89 | 29/32 |
| Taback B | 2006 | USA | QMSP | Blood | 1/33 | 0/10 | – | – | – | – |
| Van der A I | 2009 | Belgium | QMSP | FFT | 60/100 | 0/9 | – | – | – | – |
| Van der A I 1 | 2008 | Belgium | MSP | FFPET | 28/51 | 3/27 | – | – | – | – |
| Van der A I 2 | | | QMSP | FFT | 53/54 | 7/9 | – | – | – | – |
| Van der A I | 2009 | Belgium | QMSP | Blood | 15/78 | 1/19 | – | – | – | – |
| Virmani AK | 2001 | USA | MSP | Surgery | 19/45 | 3/28 | – | – | – | – |
| Wojdacz TK | 2011 | Denmark | MS-HRM | Blood | 24/180 | 13/108 | – | – | – | – |
| Zhang JJ 1 | 2007 | China | MSP | Surgery | 38/84 | 0/84 | 30/66 | 8/18 | – | – |
| Zhang JJ 2 | | | | Serum | 26/84 | 0/10 | 20/66 | 6/18 | – | – |
| Total | | | | | 1162/2483 | 96/1218 | 219/511 | 88/170 | 146/286 | 88/183 |

**Notes.**

MSP, Methylation specific PCR; QMSP, Quantitative real-time MSP; Pyro, Pyrosequencing; MS-HRM, Methylation-sensitive high-resolution melting analysis; FFPET, Formalin fixed paraffin-embedded tissue; FFT, Fresh frozen tissue; NAF, Needle aspirate fluid; MS-MLPA, Methylation specific-multiplex ligation-dependent probe amplification; M, Number of APC promoter methylated patients; N, Number of control.

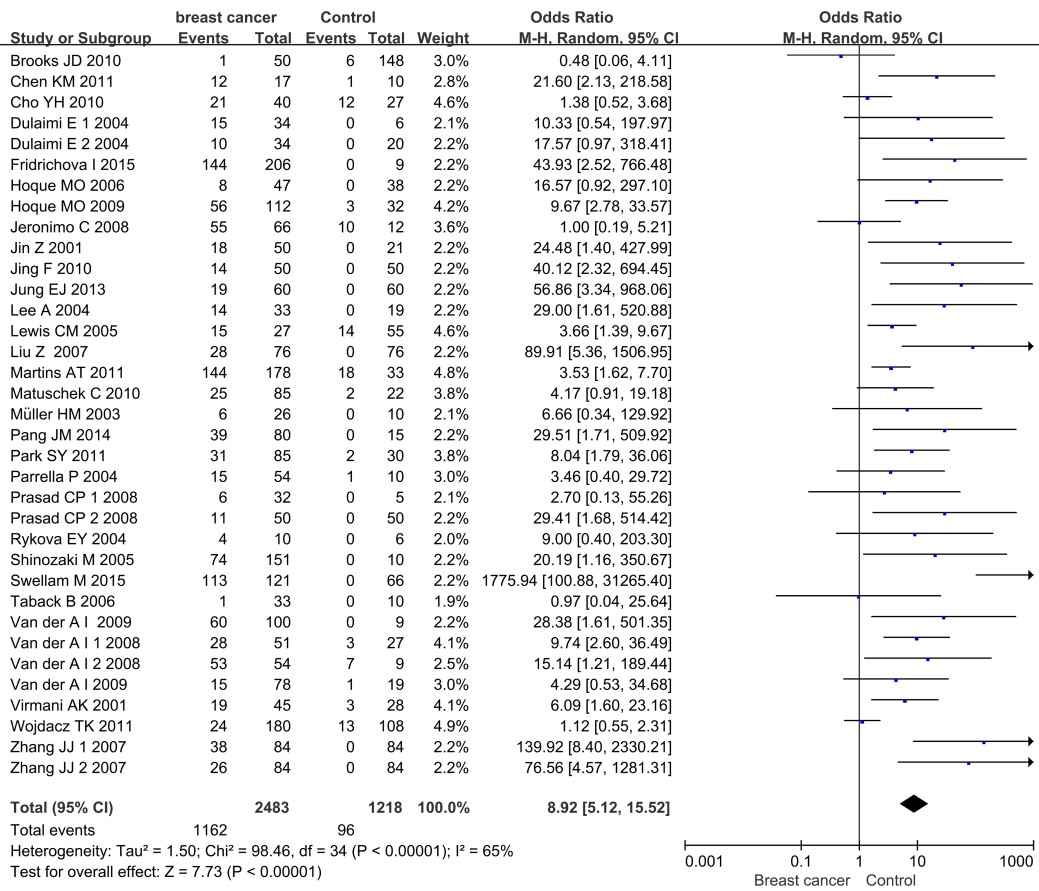

| Study or Subgroup | breast cancer Events | Total | Control Events | Total | Weight | Odds Ratio M-H, Random, 95% CI |
|---|---|---|---|---|---|---|
| Brooks JD 2010 | 1 | 50 | 6 | 148 | 3.0% | 0.48 [0.06, 4.11] |
| Chen KM 2011 | 12 | 17 | 1 | 10 | 2.8% | 21.60 [2.13, 218.58] |
| Cho YH 2010 | 21 | 40 | 12 | 27 | 4.6% | 1.38 [0.52, 3.68] |
| Dulaimi E 1 2004 | 15 | 34 | 0 | 6 | 2.1% | 10.33 [0.54, 197.97] |
| Dulaimi E 2 2004 | 10 | 34 | 0 | 20 | 2.2% | 17.57 [0.97, 318.41] |
| Fridrichova I 2015 | 144 | 206 | 0 | 9 | 2.2% | 43.93 [2.52, 766.48] |
| Hoque MO 2006 | 8 | 47 | 0 | 38 | 2.2% | 16.57 [0.92, 297.10] |
| Hoque MO 2009 | 56 | 112 | 3 | 32 | 4.2% | 9.67 [2.78, 33.57] |
| Jeronimo C 2008 | 55 | 66 | 10 | 12 | 3.6% | 1.00 [0.19, 5.21] |
| Jin Z 2001 | 18 | 50 | 0 | 21 | 2.2% | 24.48 [1.40, 427.99] |
| Jing F 2010 | 14 | 50 | 0 | 50 | 2.2% | 40.12 [2.32, 694.45] |
| Jung EJ 2013 | 19 | 60 | 0 | 60 | 2.2% | 56.86 [3.34, 968.06] |
| Lee A 2004 | 14 | 33 | 0 | 19 | 2.2% | 29.00 [1.61, 520.88] |
| Lewis CM 2005 | 15 | 27 | 14 | 55 | 4.6% | 3.66 [1.39, 9.67] |
| Liu Z 2007 | 28 | 76 | 0 | 76 | 2.2% | 89.91 [5.36, 1506.95] |
| Martins AT 2011 | 144 | 178 | 18 | 33 | 4.8% | 3.53 [1.62, 7.70] |
| Matuschek C 2010 | 25 | 85 | 2 | 22 | 3.8% | 4.17 [0.91, 19.18] |
| Müller HM 2003 | 6 | 26 | 0 | 10 | 2.1% | 6.66 [0.34, 129.92] |
| Pang JM 2014 | 39 | 80 | 0 | 15 | 2.2% | 29.51 [1.71, 509.92] |
| Park SY 2011 | 31 | 85 | 2 | 30 | 3.8% | 8.04 [1.79, 36.06] |
| Parrella P 2004 | 15 | 54 | 1 | 10 | 3.0% | 3.46 [0.40, 29.72] |
| Prasad CP 1 2008 | 6 | 32 | 0 | 5 | 2.1% | 2.70 [0.13, 55.26] |
| Prasad CP 2 2008 | 11 | 50 | 0 | 50 | 2.2% | 29.41 [1.68, 514.42] |
| Rykova EY 2004 | 4 | 10 | 0 | 6 | 2.0% | 9.00 [0.40, 203.30] |
| Shinozaki M 2005 | 74 | 151 | 0 | 10 | 2.2% | 20.19 [1.16, 350.67] |
| Swellam M 2015 | 113 | 121 | 0 | 66 | 2.2% | 1775.94 [100.88, 31265.40] |
| Taback B 2006 | 1 | 33 | 0 | 10 | 1.9% | 0.97 [0.04, 25.64] |
| Van der A I 2009 | 60 | 100 | 0 | 9 | 2.2% | 28.38 [1.61, 501.35] |
| Van der A I 1 2008 | 28 | 51 | 3 | 27 | 4.1% | 9.74 [2.60, 36.49] |
| Van der A I 2 2008 | 53 | 54 | 7 | 9 | 2.5% | 15.14 [1.21, 189.44] |
| Van der A I 2009 | 15 | 78 | 1 | 19 | 3.0% | 4.29 [0.53, 34.68] |
| Virmani AK 2001 | 19 | 45 | 3 | 28 | 4.1% | 6.09 [1.60, 23.16] |
| Wojdacz TK 2011 | 24 | 180 | 13 | 108 | 4.9% | 1.12 [0.55, 2.31] |
| Zhang JJ 1 2007 | 38 | 84 | 0 | 84 | 2.2% | 139.92 [8.40, 2330.21] |
| Zhang JJ 2 2007 | 26 | 84 | 0 | 84 | 2.2% | 76.56 [4.57, 1281.31] |
| **Total (95% CI)** | | **2483** | | **1218** | **100.0%** | **8.92 [5.12, 15.52]** |
| Total events | 1162 | | 96 | | | |

Heterogeneity: Tau² = 1.50; Chi² = 98.46, df = 34 (P < 0.00001); I² = 65%
Test for overall effect: Z = 7.73 (P < 0.00001)

**Figure 2  Forest plot of APC promoter methylation and breast cancer risk based on the random effects model.** The small squares and horizontal lines represent the OR and 95% CI of individual studies. If the 95% CI included 1, the difference in APC methylation between patients with breast cancer and controls was not significant. The centre of the diamond represents the combined treatment effect (calculated as a weighted average of individual ORs) and the horizontal tips represent the 95% CI. OR represents the odds ratio. 95% CI represents the 95% confidence interval.

(Table 2). With regard to subgroup analysis based on region, heterogeneity in Asian subgroups disappeared completely ($I^2 = 0\%$) and the pooled OR value was 24.48 [10.94, 54.74]. The $I^2$ value representing heterogeneity in the European and North American subgroups decreased by 50% and 42%, compared with the overall $I^2$ value. Furthermore, their OR values also decreased to 4.63 [2.44, 8.78] and 3.79 [1.70, 8.44]. In the African subgroup, the OR was 172.05 [1.76, 16792.96] with higher heterogeneity ($I^2 = 80\%$) due to the small subset containing only 2 studies. These results indicated that the heterogeneity might result from different regions and APC methylation was remarkably related to increased BC risk without geographical restrictions. For the subgroup analyses based on sample types, the blood or serum group (OR = 9.44, 95% CI [2.56–34.83]) made the largest contribution to the heterogeneity ($I^2 = 78\%$). In the tissue subgroup, the OR was 9.93 [5.10, 19.34] with lower heterogeneity ($I^2 = 50\%$). Moreover, heterogeneity in the NAF subgroups could be ignored ($I^2 = 6\%$). These results confirmed the stable association between APC methylation and BC risk in different sample types. For studies based on the methods used to detect

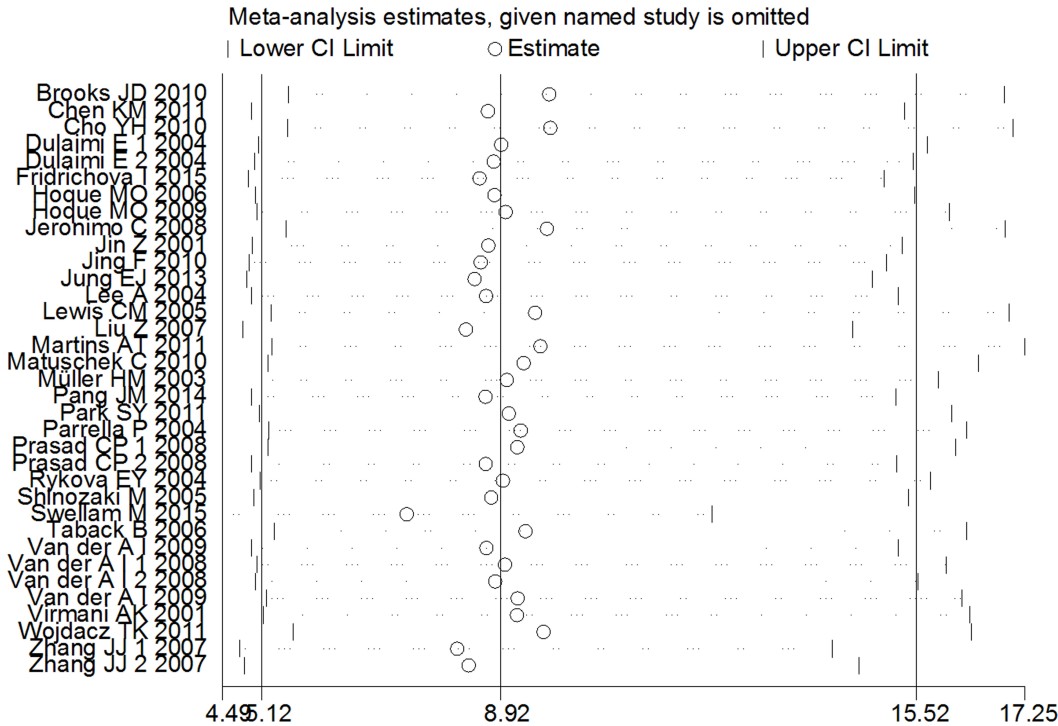

**Figure 3** **Sensitive analysis of pooled OR based on the random effects model.** The results were calculated by omitting each study in turn. The circles represent the individual studies in this meta-analysis. The two ends of the dotted lines represent the 95% CI. OR represents the odds ratio. 95% CI represents the 95% confidence interval.

the methylation of the APC promoter, the combined OR value was 18.18 for MSP (95% CI [7.96–41.52]), 3.93 for QMSP (95% CI [1.78–8.69]), 3.29 for MethyLight (95% CI [1.27–8.52]) and 31.81 for MS-MLPA (95% CI [5.30–191.06]). Heterogeneity in the QMSP ($I^2 = 36\%$) and MS-MLPA ($I^2 = 0\%$) subgroups was far lower than that of the MethyLight and MS-HRM subgroups ($I^2 = 83\%$).

To assess the association between APC methylation and tumor stage, 11 studies comprising 681 BC patients were pooled to calculate the OR. The results showed that the frequency of APC promoter methylation was significantly lower in early-stage patients than in late-stage patients (OR = 0.62, 95% CI [0.42–0.93], $I^2 = 34\%$). Meanwhile, the OR of 8 studies revealed that the association between APC methylation and tumor grade was not statistically significant (OR = 0.78, 95% CI [0.51–1.21], $I^2 = 0\%$).

## Publication bias

We used the funnel plot, Begg's test and Egger's test to evaluate the degree of publication bias. The shape of the funnel plot had no obvious asymmetry (Fig. 4A). Moreover, Begg's test ($Pr > |z| = 0.239 > 0.05$) suggested no significant publication bias (Fig. 4B). Interestingly, Egger's test revealed evident statistical proof for the existence of publication bias ($p > |t| = 0.000 < 0.05$). Therefore, we carried out trim and fill analysis to identify and revise the bias. As shown in Fig. 4C, 12 adjusted studies were added to the initial meta-analysis. The corrected OR was still highly significant for the association between APC

**Table 2 Subgroup analysis for the relationship between APC promoter methylation and breast cancer.**

| Subgroup | No | BC M/N | Control M/N | OR (95% CI) | Heterogeneity test | | |
|---|---|---|---|---|---|---|---|
| | | | | | $I^2$ | $p$ | $Chi^2$ |
| **Sample types** | | | | | | | |
| Tissue | 19 | 731/1397 | 42/480 | 9.93 [5.10, 19.34] | 50% | 0.006 | 36.34 |
| Blood or Serum | 13 | 258/848 | 22/631 | 9.44 [2.56, 34.83] | 78% | <0.00001 | 55.34 |
| NAF | 3 | 173/238 | 32/107 | 3.95 [2.10, 7.42] | 6% | 0.34 | 2.13 |
| **Region** | | | | | | | |
| Asia | 10 | 205/604 | 2/479 | 24.48 [10.94, 54.74] | 0% | 0.53 | 8.07 |
| Europe | 13 | 629/1200 | 58/306 | 4.63 [2.44, 8.78] | 50% | 0.02 | 24.18 |
| North America | 9 | 168/430 | 36/314 | 3.79 [1.70, 8.44] | 42% | 0.09 | 13.76 |
| Africa | 2 | 121/168 | 0/104 | 172.05 [1.76, 16792.96] | 80% | 0.02 | 5.07 |
| Oceania | 1 | 39/80 | 0/15 | 29.51 [1.71, 509.92] | NA | NA | NA |
| **Methods** | | | | | | | |
| MSP | 17 | 448/986 | 21/617 | 18.18 [7.96, 41.52] | 54% | 0.004 | 35.03 |
| QMSP | 9 | 393/718 | 45/310 | 3.93 [1.78, 8.69] | 39% | 0.11 | 13.20 |
| MethyLight | 4 | 83/236 | 16/89 | 3.29 [1.27, 8.52] | 36% | 0.20 | 4.66 |
| MS-MLPA | 2 | 31/77 | 1/70 | 31.81 [5.30, 191.06] | 0% | 0.58 | 0.31 |
| MS-HRM | 2 | 63/260 | 13/123 | 4.49 [0.14, 146.62] | 83% | 0.02 | 5.76 |
| Pyro | 1 | 144/206 | 0/9 | 43.93 [2.52, 766.48] | NA | NA | NA |

**Notes.**

NAF, Needle aspirate fluid; MSP, Methylation specific PCR; QMSP, Quantitative real-time MSP; Pyro, Pyrosequencing; MS-MLPA, Methylation specific-multiplex ligation-dependent probe amplification; MS-HRM, Methylation-sensitive high-resolution melting analysis; NA, Not available; M, Number of APC promoter methylated patients; N, Number of control.

methylation and BC risk (OR = 1.444, 95% CI [1.081–1.965]), further proving the stability of our meta-analysis.

## DISCUSSION

To the best of our knowledge, this is the first meta-analysis to systematically evaluate the association between APC promoter methylation and BC pathogenesis. BC is a significant clinical and public health problem and is mainly attributed to epigenetic and genetic changes. Epigenetic alternation involving DNA methylation is a relatively early event that serves as a tumor molecular biomarker candidate in BC and can be detected in all pathological tumor stages. The APC gene is considered to be a tumor suppressor gene, and the silencing of its expression may result in cell-to-cell adhesion disorders and the disruption of the Wnt signaling pathway. APC methylation, a contributing factor to the absence of APC expression, is often linked to $\beta$-catenin accumulation and TCF/LEF-induced transcription (*Klarmann, Decker & Farrar*, *2008*). Numerous studies have reported that APC methylation is highly specific for BC and can be used as a biomarker in the diagnosis of BC (*Dumitrescu*, *2012*; *Van der Auwera et al.*, *2008*). *Zhang, Li & Lang* (*2015*) found that $\beta$-catenin overexpression was significantly associated with an unfavourable prognosis in patients with breast cancer. However, the role of APC methylation in BC pathogenesis remains controversial.

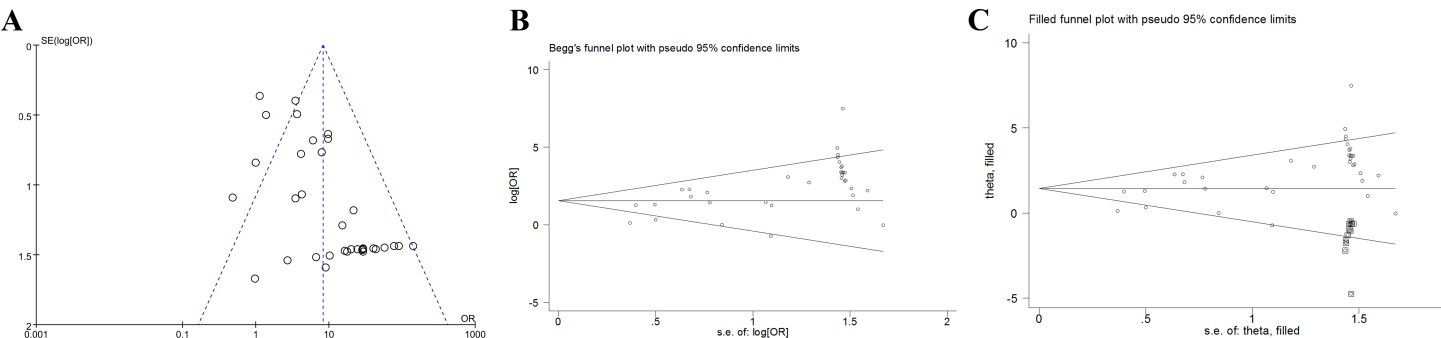

**Figure 4** **Publication bias analysis.** (A) The funnel plot of APC methylation and breast cancer risk. The log of OR against the standard error of the log of the OR was plotted in this graph.(B) The Begg's plot of APC methylation and breast cancer risk. The circles represent the individual studies in this meta-analysis. The line in the centre represents the pooled OR. (C) The Begg's plot of publication bias after trim-and-fill analysis. The circles represent the included studies. The diamonds represent the presumed missing studies. OR represents the odds ratio.

To resolve these contradictory results, we gathered relevant studies and carried out this meta-analysis using systematic statistical methods. Herein, we included a total of 35 studies with 2,483 cases and 1,218 controls published from 2001 to 2016. Our results based on the pooled OR revealed that the level of APC methylation was observably higher in BC patients compared to cancer-free controls, which indicated that APC methylation could serve as a potential biomarker for BC diagnosis, regardless of the various sample types detected, APC methylation detection methods applied and cases employed in different regions.

Then, we conducted subgroup analysis to identify the sources of the heterogeneity and found that various sample types, methylation detection methods and cases employed in different regions all contributed to the heterogeneity. In subgroup analysis based on sample types, the results showed that APC methylation was significantly related to BC pathogenesis, whether in tissue, blood or serum and NAF. Cell-free DNA in serum and plasma, which mostly originates from tumor cell degradation, can be collected and examined for epigenetic alterations with various malignancies (*Anker et al.*, *1999*). The sample materials including blood or serum, used for extracting DNA are often stored for different time periods which will produce false positives and false negatives. Thus, blood samples should be examined as rapidly as possible after being collected. Therefore, the accuracy of cell-free DNA largely depends on the standardized storage conditions. NAF is a rapid, minimally invasive and cheap diagnostic means with high sensitivity. The accuracy of NAF mainly relies on the experience of the cytopathologist which may result in an increasing trend for false negatives (*Jeronimo et al.*, *2003*). In subgroup analysis based on methylation detection methods, significant associations were observed when examined using MSP, QMSP, MethyLight and MS-MLPA, except for MS-HRM. Among these, the pooled OR derived from studies using MS-MLPA was the maximum with no heterogeneity. The diagnostic accuracy of MS-MLPA was not affected by sample types (*Park et al.*, *2011a*). Cut-off values and primers based on different CPG islands which were used in different studies, contributed to the heterogeneity of other methods. In subgroup analysis based on different regions, APC methylation was significantly correlated with BC patients in all included regions. The results indicated that although the genetic factors, environments and life styles were totally different, the

correlation was still strong and stable. Therefore, an appropriate APC methylation detection method considering the regions and sample types employed is essential for routine clinical diagnosis. Additionally, we found that the status of APC methylation increased notably in late-stage patients compared with early-stage ones, which indicated that APC methylation might be closely related to the malignant evolution of BC.

As mentioned above, *Wojdacz et al.* (*2011b*) examined the use of methylation biomarkers as screening tools for BC diagnosis. They found no significant difference in the frequency between 180 BC patients and 108 healthy controls and a weak association between APC methylation and BC pathogenesis. This discrepancy mainly resulted from the methylation detection method. They used MS-HRM which may yield heterogeneous methylation values derived from the primer and cut-off values, and it tended to produce a lower evaluation of methylation when applying less methylated samples (*Migheli et al.*, *2013*).

Surprisingly, only Egger's linear regression showed an obvious publication bias other than Begg's test and funnel plots. *Egger et al.* (*1997*) suggested that Egger's test was more sensitive than Begg's test. The publication bias mainly resulted from the inclusion criteria. Only full-text published studies were collected in this meta-analysis. Therefore, unpublished studies and conference abstracts were not included. Additionally, other study characteristics including the source of funding and prevailing theories at the time of publication, can contribute to publication bias. However, we included a large number of BC patients ($n = 2,483$) to ensure the reliability of the meta-analysis and minimize the potential publication bias.

Although the meta-analysis indeed confirmed the significance of a correlation between APC methylation and BC pathogenesis, several limitations should be considered. First, the sample sizes used in several studies were small, which may have increased the risk of publication bias and limited the results of the meta-analysis. Second, the quality of the selected studies varied, as we included high-quality and low-quality studies. Therefore, heterogeneity likely existed. Third, the cut-off points of APC methylation and the primers based on CPG islands were difficult to unify. Thus, we were unable to calculate the pooled sensitivity and specificity of APC methylation.

In conclusion, the results of our meta-analysis highlight the clinical significance and scientific value of APC promoter methylation in the diagnosis of BC. Consequently, APC methylation is a potential biomarker for monitoring BC development. However, given the limitations listed above, high-quality studies with large-scale and consistent standards should be carried out. The guidelines for the reporting of tumor marker studies recommended by the National Cancer Institute are necessary for adaptation to high-quality studies (*McShane et al.*, *2005*).

### Funding

This study was supported by grants from the National Natural Science Foundation of China (No. 31400699) and the Natural Science Foundation of Fujian Province of China (No. 2014J01142). The funders had no role in study design, data collection and analysis, decision to publish, or preparation of the manuscript.

## Grant Disclosures

The following grant information was disclosed by the authors:
National Natural Science Foundation of China: 31400699.
Natural Science Foundation of Fujian Province of China: 2014J01142.

## Competing Interests

The authors declare there are no competing interests.

## Author Contributions

- Dan Zhou conceived and designed the experiments, performed the experiments, wrote the paper, prepared figures and/or tables.
- Weiwei Tang performed the experiments, prepared figures and/or tables.
- Wenyi Wang analyzed the data.
- Xiaoyan Pan contributed reagents/materials/analysis tools.
- Han-Xiang An and Yun Zhang conceived and designed the experiments, reviewed drafts of the paper.

## Data Availability

The raw data has been supplied as a Files S1 and S2.

## Supplemental Information

Supplemental information for this article can be found online at http://dx.doi.org/10.7717/peerj.2203#supplemental-information.

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
