# Peer review of "Association between aberrant APC promoter methylation and breast cancer pathogenesis: a meta-analysis of 35 observational studies"

_PeerJ, doi:10.7717/peerj.2203_

## Round 0.1 · original submission · Major Revisions

You work has been reviewed by three topic experts who have raised several relevant issues. Please address these for further consideration for publication in this journal.

·

Basic reporting

Authors for this paper have carried out a Meta-analysis to assess the correlation between APC methylation and Breast Cancer risk. The question at the focus of this meta-analysis is whether APC promoter methylation is associated with increased BC risk in the presence of contradictory results. Objective, study design and statistical tools for this study has been clearly stated and it is clinically relevant but there are some significant problems with the sample size and exclusion criteria that reduce the significance of this analysis. Following are the major concerns:
1. The authors have not conducted a comprehensive search for unpublished studies related to the research question. Also the reasons to exclude the unpublished studies are not clear as the analysis should include published and unpublished results to avoid publication bias..
2. Though the publication bias has been addressed through evaluation methods such as funnel plot or Begg’s test, the heterogeneity dangers of combining results from high quality vs low quality studies cannot be ignored. Nevertheless, if there is the existence of heterogeneity, then the authors should emphasize the need for high quality study and provide recommendations for optimal study design.
3. It has been stated that the relevant data were extracted independently by two authors but disagreements are not listed between authors and how they were resolved.

Experimental design

1. The rational to exclude the data based on reviews, animal models, case report or cell line studies have not been given which also leads to reduced sample size.
2. 17 out of 35 total studies have used Methylation specific PCR; which is prone to false positive. Therefore the studies included in the analysis should have the right pool of different methods.

Validity of the findings

1. Appropriate statistical methods have been used to combine results but the sample size is too small to yield the statistical power. Specially the large data set with high quality is only few.
2. There are 2482 BC patients but only 1212 controls which weaken the comparison.

Additional comments

1. Figure 1-4 does not have detailed legends; therefore they are not self explanatory. Appropriate abbreviations and details should be added.
2. There is couple of typographical errors in the publication for example:
Line 260 funnl replace it with funnel
Figure 3 title Seneitiive change it to Sensitivity
Frost to Forest plot in the labelling of the figure at submission

·

Basic reporting

Author must describe more about "Forest Plot" and "Begg’s plot" in the methods.

Experimental design

Even if this study is good for Wnt signaling effect on breast cancer, but is limited to APC methylation and doesn’t include other crucial molecule which has been shown to affect Wnt signaling like TCF, GSK3β etc.

Validity of the findings

No Comments

Additional comments

Adenomatous polyposis coli (APC) is widely known as an antagonist of the Wnt signaling pathway via inactivating β-catenin, and APC methylation contributes a lot to the predisposition of breast cancer. Authors have concluded based on the Meta-analysis that the frequency of APC methylation is significantly higher in BC cases than controls under a random effect model. Also, Frequency of APC methylation was significant lower in early-stage BC patients than late stage ones. It’s a good Meta-analysis but need to address following Major concern.
1) A Meta-analysis on APC methylation and Wnt signaling downstream target genes must be done to show that the APC methylation is correlated with activation of Wnt signaling in different stages of breast cancer.
2) A Meta-analysis on Breast cancer survival studies and APC methylation or Wnt signaling (β-catenin) must be done to get a better picture on the relevance of APC methylation in Breast Cancer

Reviewer 3 ·

Basic reporting

The manuscript sent for review is a meta analysis of the role of APCgene promoter hypermmethylation as a biomarker for breast cancer and its role in understanding the pathogenesis of breast cancer. the data from this analysis would benefit researchers working in this area and to plan further studies.

Experimental design

The authors have used the various search engines available to scan the literature and have finally selected 35 publications for detailed analysis.Literature survey is adequate.

Validity of the findings

The data collected has been analysed systematically using appropriate statistical tools and the data has been presented as figures.

Additional comments

The figure captions can be better and in Fig 4 B the title has been given as Beggs plot of GST methylation and breast cancer.
The introduction needs correction as there are grammatical errors.
The discussion has not been written well and needs a complete rewriting.. The authors are encouraged to take professional help to improve the manuscript to avoid grammatical errors and to improve the overall presentation.

The manuscript may be accepted for publication after revision.

---

## Round 0.2 · accepted · Accept

Thank you for your revision that has suitably addressed the concerns raised. Congratulation!

·

Basic reporting

Authors have incorporated all the changes

Experimental design

Authors have incorporated all the changes

Validity of the findings

No Comments

Additional comments

Authors have incorporated all the changes